# Comparative Analysis of Dropout and Student Permanence in Rural Higher Education

**Alfredo Guzmán** [1,*], **Sandra Barragán** [2] **and Favio Cala-Vitery** [2]

1 Center of Research of Asturias (Centro de Investigación de Asturias), Corporación Universitaria de Asturias, Bogota 110221, Colombia
2 Faculty of Natural Sciences and Engineering, Universidad de Bogota Jorge Tadeo Lozano, Bogota 111711, Colombia; sandra.barragan@utadeo.edu.co (S.B.); favio.cala@utadeo.edu.co (F.C.-V.)
* Correspondence: alfredo.guzman@asturias.edu.co; Tel.: +57-321-254-0363

**Abstract:** The growing dropout and low permanence of rural students in higher education has become a central problem in the education system, both affecting the quality conditions of training programmes and preventing the materialisation of the benefits that achieving this educational level entails for society. However, the study of these events in rural populations is scarce, resulting in an inadequate treatment of dropout and, consequently, the impossibility of consolidating student permanence. Thus, the aim of this article is to identify which individual, academic, socio-economic, and institutional variables influence the dropout and the retention of the rural student population in higher education. To achieve this purpose, a cross-sectional study was defined. The sample used was a non-probabilistic sample with an n of 269 rural Colombian students who were administered a self-report questionnaire that assessed 59 variables. Data analysis was based on means comparison and cluster modelling. The results show that dropout and permanence in rural students is related to the educational level of the father, family and work obligations, the need to move from their place of residence, the academic average in higher education, satisfaction with the choice of programme, communication with the institution, and the attention of teachers, among other things.

**Keywords:** educational quality; higher education; dropout; permanence; public policies; institutions; modelling





## 1. Introduction

The term quality is widely used in higher education systems worldwide to ensure excellence at both the institutional and training programme levels [1–3]. In this sense, it is necessary to recognise that, before the 1980s, quality in higher education was an internal matter for Higher Education Institutions (HEIs); however, after the 1980s, quality at the level of training became a matter of public policy, making quality assessment an internal activity of HEIs, as well as an external activity of interest to states [4,5]. Thus, in the quality assessment exercise, various standards which allow us to understand the current state of substantive functions (teaching, research, and relations with the external sector), as well as those complementary to these functions, have been generated [6].

In this scenario, there are multiple indicators that evaluate the quality of the educational system, HEIs and training programmes; however, the student dropout rate and its counterpart, the permanence rate, have become one of the main indicators [7–9], since they allow us to identify whether training programmes manage to provide society with professionals who meet the diverse demands that society generates on a continuous basis [10]. Hence, if an HEI, a training programme or an education system does not rank below the average dropout rate at national or global level, it is considered to be of low quality, leading to intervention through the development of institutional policies and public policies to avoid this scenario.

The intervention generated by both HEIs and the State is not only related to the outcome of the dropout or permanence rate, but also to the effects that these educational events bring to society, by limiting or achieving the materialisation of the benefits of higher education (e.g., higher income, increased productivity and better security rates, etc.) [11]. This makes both dropout and permanence at the educational level a matter of interest for the academic community, as well as for policy makers.

In accordance with the above, many studies have sought to establish the variables that explain the materialisation of these events both in HEIs [11–18] and in the education system [19–24]. However, their study still lacks multiple perspectives, generating indications that HEI and state policies have not been effective, thus high dropout rates and low permanence in education systems persist. An illustration of this is often the situation in OECD nations, where in 2018 the dropout rate was near to 64.5%, or, in the case of Latin America, the dropout rate was close to 54% [25]. In addition, the dropout rate since the beginning of the COVID-19 health crisis has increased, especially among vulnerable student populations (e.g., those displaced by conflict, Afro-descendants and rural populations, among others) [26].

The study of dropout and permanence in rural populations is placed within the framework of perspectives that have been little addressed by both academics and education policy decision makers [11,25]. As a result, both HEIs and states have dealt with dropouts in this student population with generic strategies that are applied equally to all types of students, without considering the individual, academic and socio-economic aspects of the students and the institutions in which they study. This has led to an increase in the dropout rate and a low permanence rate, thus affecting the quality of the training programmes offered in these areas [27]. Hence, it is necessary to evaluate the institutional and public policies that should be implemented in the rural student population to prevent and mitigate the event of dropout, in order to achieve the permanence of this type of student in the education system. In the analysis of this problem for rural students, the need arises to know what variables influence the decision to abandon or remain in the educational process.

Thus, the aim of this article is to identify which individual, academic, socio-economic and institutional variables influence the dropout and retention of rural students in higher education. The Colombian education system was selected for this study because most of the previous research on dropout or permanence of rural students has been carried out in developed countries [28–32], and not in contexts of social disparity as marked as the Colombian case, where rural areas have been characterised by violence and conflict by various armed actors, which has led to marginalisation, inequality in the income of the population, regional differences and various social tensions [11]. Hence, this analysis in the Colombian rural population, as an added value, allows us to understand what other variables influence dropout or permanence, providing new perspectives for the academic community, as well as for public policy and HEI decision makers.

This article is structured in four sections. The first presents the theoretical framework and contextualisation of dropout and retention in Colombia and the studies developed; the second contains the methodology that allowed the fulfilment of the objective; the third shows the results; the fourth discusses the main findings and offers the conclusions of this study.

## 2. Theoretical Background

### 2.1. Dropout and Permanence in Higher Education

Dropout as an event that affects education systems does not have a unique meaning, being the result of the different actors involved in its study, such as researchers, HEIs, states and social organisations, among others [25]. That said, the literature tends to conglomerate definitions of dropout into two main groups. The first group is a compilation of conceptualisations derived from the academic study of dropout; the second group is operational, established by states in the framework of education systems to facilitate the quantification of the event [11,33].

In this sense, the present article is framed within the first group, which allows for the analysis of multiple variables that can lead to the early termination of a student's academic studies. Thus, dropout is defined as "the cessation of the relationship between the student and the training programme leading to a higher education qualification before the qualification is recognised. An event of a complex, multidimensional and systemic nature, which can be understood as cause or effect, failure or reorientation of a training process, compulsory choice, or response, or as an indicator of the quality of the education system" [34] p. 6. The use of this meaning of dropout permits the integration of perspectives on the study of this event. In the case of permanence, there is a greater consensus regarding its conceptualisation, which is understood as "the permanent initiative of HEIs to generate strategies to strengthen institutional capacity, which contribute to reducing drop-out rates". It is also an important element in the elaboration of "the institutional educational plan" [35] p. 194.

In view of the various actors involved in the research of dropout and permanence in higher education, the multidisciplinary orientation in its study stands out [35]. This has led to the examination of illustrative factors, both innate and external to the student, which can be categorised as: individual, socio-economic, academic and institutional. This categorisation has been used in previous studies, such as those of Fonseca and García [36], Barragán and González [16,24], Donoso and Schiefelbein [37] and Guzmán et al. [11,25], among others. In addition, states have made use of this categorisation to define public policies to prevent and mitigate dropout at the educational level, as is the case in Colombia [38]. Figure 1 presents the dropout model based on the categorisation of variables; each cluster of variables is hereafter referred to as a determinant. It should be noted that the variables of one determinant have the capacity to relate to and influence one another. These same determinants can, in fact, also explain permanence in higher education.

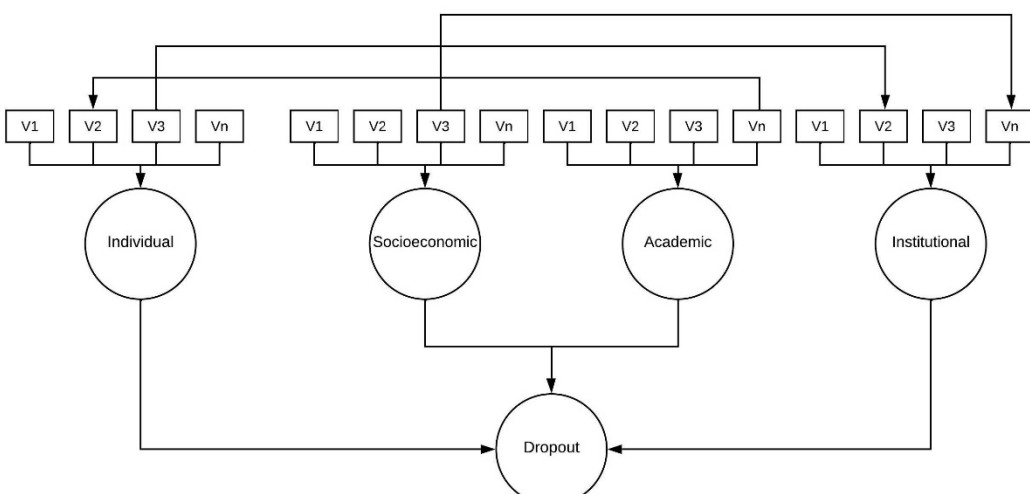

**Figure 1.** Conceptual model of determinants of dropout. Note: Each determinant groups n variables v1, v2, . . . , vn. Reprinted with permission of Guzmám et al. [21]. Copyright 2021 Frontiers.

The individual determinant explains the characteristics related to the student and his or her individual environment that specifically affect the choice of whether to leave higher education without completing it or to remain in it [38]. In higher education, the individual determinant variables have been widely debated, as several research studies have largely attributed them to the materialisation of the dropout event [39]. An example of this was evidenced in the study by Georg [40], who found that 95% of dropouts from German HEIs were explained by the characteristics of the individual at the time of entry into the institution. The socio-economic determinant refers to the variables of the social and economic environment that affect the student and his or her family and that directly or indirectly affect dropout or permanence [38]. Previous studies have been divergent,

since some research has indicated that this type of variable does not influence dropout or permanence in higher education [41,42] and others have highlighted the influence of these variables on student completion of the educational process [43–45].

The academic determinant relates to the achievement of learning outcomes, the advancement of proficiency, student performance and other components that impact the management of instruction and learning at all levels of instruction [38]. In general, the findings of previous studies identify that the variables of this determinant have a great impact on student dropout and permanence in higher education, especially because of the demands of the educational level, as identified by Heidrich et al. [15], Choi and Kim [46], as well as Stewart et al. [47,48]. Transition to higher education [14] and student perspectives (e.g., self-efficacy and self-management) [48] are closely related to dropout at the educational level.

Finally, the institutional determinant explains those characteristics of HEIs that allow for the correct development of the educational process [38]. Previous research has found that the high levels of attrition and retention in HEIs are related to their size in terms of number of students, the quality of the training programmes, programmes for permanence and timely graduation (P&GOs) and administrative processes [46,49,50].

### 2.2. Context of Dropout and Permanence in Higher Education in Colombia

Student desertion and permanence as indicators of quality in higher education in Colombia began to be of interest to the state in 2003, with the implementation of the first strategies for the prevention and mitigation of desertion and the achievement of permanence [51]. As a result of these initial efforts, there was a need to expand the study of student dropout and permanence through accurate and reliable information, and the National Education Ministry (NEM) consolidated both the state information system SPADIES (Sistema para la Prevención de la Deserción de la Educación Superior in Spanish) and various public policies. Simultaneously, the national academic community became interested in the study of these educational events.

In the case of the state, public policies aimed at preventing and mitigating dropout have been designed and implemented jointly with HEIs. Thus, the state has taken on the role of funder for students, providing educational credits and scholarships [11,52]; in addition, HEIs have focused on strengthening competences, as well as developing Early Warning Systems (SAT in Spanish) and P&GOs to identify and support students at risk of not completing their educational programme [53]. As a result of these efforts and according to SPADIES data [54], in the first semester of 2021, the dropout rate of the system was 7.6%, while for the second semester it was 12.8%. While the dropout rates, as presented in Figure 2, are below those observed in the Latin American and OECD region, when analysing the situation of training institutions and programmes located in rural areas, the reality is different.

Thus, for the year 2016, it was estimated by the NEM that the dropout rate by cohort in rural areas was close to 50%, both for technical and technological levels, as well as for the university level [55]. However, at the national level, in these areas, the dropout rate varies between departments or states. An example of this is the departments of Chocó, La Guajira and Putumayo, where the dropout rates for technical and technological programmes were 91.3%, 73% and 71.2%, respectively, while for university programmes the departments with the highest dropout rates were Putumayo, La Guajira and Arauca, with rates of 80.2% and 55.6% for the latter two [56]. In both cases, these departments are characterised by high levels of social disparity [57].

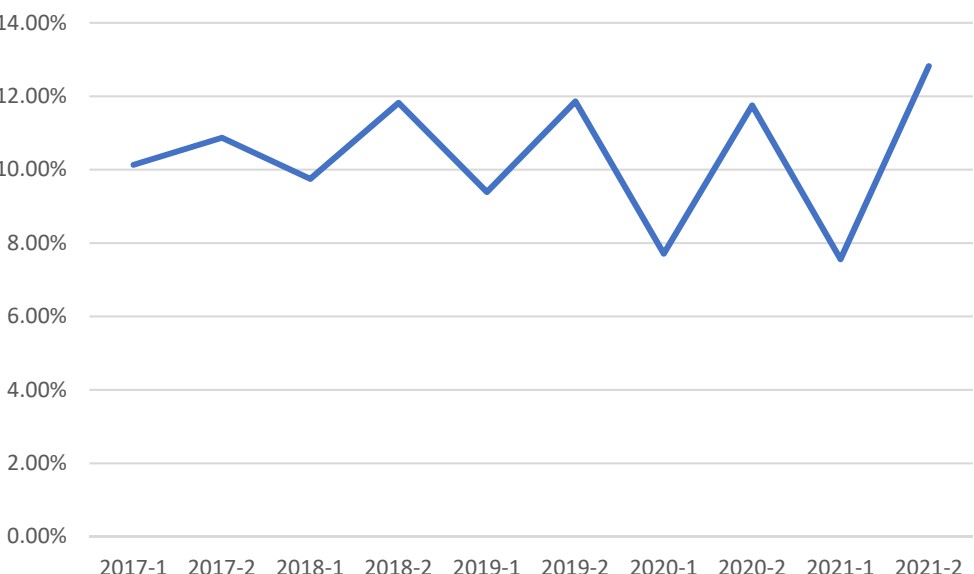

**Figure 2.** Dropout rate in the Colombian higher education system for the years 2017 to 2021. Adapted with permission of [54]. Copyright 2022 SPADIES.

However, public policies implemented by the state to prevent and mitigate the effects of dropout and achieve permanence in the rural student population are characterised by being non-differential, based on the financing and support of HEIs [11]. This is due to a certain extent to the lack of information on what is happening in the education system in these areas, a situation that is not exclusive to Colombia, but which is also present in other countries, as Byun et al. [27] and Castleman and Meyer [30] state, since government information systems do not incorporate the rurality variable, and there is a lack of academic interest in researching dropout and permanence in this student population [25].

Having said that, in the case of the research developed in Colombia on the desertion and retention of rural students in higher education, only three studies have been carried out. The first was carried out by Rueda et al. [58], who determined that rural students who are at greater risk of dropping out were characterised by a low level of maladjustment or adaptation to university life, as well as belonging to single parent families and with severe or moderate family dysfunction. The second, which was developed in virtual education training programmes, established that academic variables do not influence the events of desertion or permanence, whereas conjugal status (related to family commitments), age, social status, work commitments, parents' education level and sort of work, the student's pay and sort of work relationship, as well as the number of individuals who depend on the family's income, do influence the events of desertion or permanence [11]. Finally, the third assessed the potential of student dropout in higher education to widen social gaps in rural Colombia, as it is the student and his or her family who bear the greatest costs associated with these events [21].

## 3. Methodology

To fulfil the objective of this article, which is to identify the individual, academic, socioeconomic and institutional variables that influence the dropout and permanence of rural students in higher education, a quantitative cross-sectional study was carried out, following the parameters established by Sedgwick [59] and Cvetković-Vega et al. [60]. The sample, instruments and explanatory variables are described below, as well as the data analysis and modelling.

### 3.1. Sample

For the present study, a non-probabilistic, non-intentional sampling was defined, so that the selection of information-rich cases was sought, using Patton [61] as a theoretical

reference for the selection criteria of the participants, of which three were established. The first is to be linked to an undergraduate training programme (technician, technologist or undergraduate); the second is to express the intention to drop out or remain in the training programme; and the third is to be in or come from a rural area. As this was a non-probabilistic sample, support was requested from the RUPPEGO network (Red Universitaria por la Permanencia Estudiantil in Spanish), which groups 37 HEIs at the national level. Based on the above, the final sample was 269 rural students, of whom 131 reported having the intention to drop out and 138 to remain in education. The sample size was similar to that of previous studies, such as those developed by Guzmán et al. [11], Contreras [62] and Oasi et al. [63]. Although there is parity between the sample of the present study and others developed previously, it is clarified that the results presented here should not be generalised, due to the nature of the type of sampling selected. Table 1 presents the general characteristics of the study sample.

**Table 1.** General characteristics of study participants.

| Characteristics | Result |
| --- | --- |
| Gender | Male: 40.89% |
| | Female: 59.11% |
| Age | 17–20: 6.31%<br>21–24: 14.12%<br>25–28: 16.72%<br>29–32: 13.75%<br>+33 years: 49.07% |
| Current semester | 1: 27.14%<br>2: 11.52%<br>3: 7.81%<br>4: 10.78%<br>5: 9.29%<br>6: 11.15%<br>+7 semester: 21.93% |
| Family income level * | COP0 to COP500,000: 12.63%<br>COP500,001 to COP1,000,000: 27.13%<br>COP1,000,001 to COP1,500,000: 25.65%<br>COP1,500,001 to COP2,000,000: 18.21%<br>COP2,000,001 to COP2,500,000: 5.94%<br>COP2,500,001 pesos or more: 10.40% |

* The values shown are in Colombian pesos. For conversion purposes, 1 USD is equivalent to 3950 COP, as of 30 March 2022.

The Study Participants Living in the Departments of Antioquia, Bolívar, Cundinamarca, Caquetá, Nariño, Guajira and Chocó.

### 3.2. Instruments and Explanatory Variables

An online self-reporting questionnaire was used to collect the data. The questionnaire was developed ad hoc based on the theoretical models proposed by Tinto [64], Barragán and González [16], Kemper et al. [65], Guzmán et al. [25], Segovia-García et al. [9], Heublein et al. [66], as well as Aina et al. [67], among others. The questionnaire was divided into six sections. The first sought to obtain informed consent and authorisation to participate in the study, and collected data from the students on their intention to drop out or remain in the training programme, as well as the type of programme they are studying; the second collected data on the variables of the individual determinant; the third focused on the socio-economic determinant variables; the fourth assessed the academic determinant variables; the fifth was related to the institutional type variables; and the last sought to confirm the student's rurality condition. Table 2 shows the variables analysed by the questionnaire and their theoretical contribution. To support this, the advances in the field of study related to

rurality and in those variables not dealt with in previous research on the rural student population were taken, as well as the basis of the research developed on other types of students. Similarly, Table A1 in the Appendix A presents the instrument and the coding of the study variables.

**Table 2.** Explanatory variables assessed.

| Determinant | Variable | Theoretical References |
|---|---|---|
| Individual | Age | [28,68] |
| | Gender | [28,69] |
| | Work obligations | [28,70,71] |
| | Family obligations | [28,70] |
| | Marital Status * | [24] |
| | Parents' level of education | [72] |
| | Student psychological traits | [30,70,72] |
| Socio-economic | Type of dwelling * | [18] |
| | Stratum | [11] |
| | Access to public services * | [9] |
| | State benefits * | [9] |
| | Family income | [29,73] |
| | Methods of financing studies * | [12] |
| Academic | Type of school graduated from | [68,74] |
| | Dropout from other previous academic programmes * | [14] |
| | Entry time to higher education * | [14] |
| | Number of subjects taken * | [14] |
| | Academic behaviour, attitudes and self-perceptions | [70,72] |
| Institutional | Use of university welfare programmes | [70,75] |
| | Communication with the HEI | [17] |
| | Attention of the HEI administrative staff * | [14] |
| | Technologies used by the HEI related to the training programme | [76] |
| | Teaching role * | [17] |
| | Participation in extracurricular activities * | [22] |

* Corresponds to variables not addressed in the literature on higher education dropout among rural students.

In relation to the reliability of the instrument, an internal consistency analysis was carried out for each of the four determinants assessed, using the Cronbach's Alpha statistic ($\alpha$). In this way, $\alpha$ was considered moderate in the event that its esteem was between 0.40 and 0.60, satisfactory between 0.60 and 0.80, and high when it was above 0.80 [77]. Additionally, it was decided to eliminate the explanatory variable in the determinant in question if this improved the value of $\alpha$. Table 3 presents the reliability of the applied questionnaire.

Based on the results in Table 3 and to ensure the reliability of the questionnaire, variables that improve the value of $\alpha$ were eliminated from the analysis, both for the selection of statistical tests to be used and for the analysis of the results. Thus, in the case of the individual determinant, variable I1 was eliminated so that $\alpha$ was considered moderate (0.58); for the socio-economic determinant S1, S11 and S14 were eliminated so that $\alpha$ was acceptable (0.60); for the academic determinant A2, A4 and A15 were eliminated so that $\alpha$ was acceptable (0.701); and, finally, for the institutional determinant, IES1 and IES8 were eliminated so that $\alpha$ was acceptable (0.781).

**Table 3.** Reliability of the self-reporting questionnaire.

| Determinant | Code | $\alpha$ | $\alpha$-SE ** |
|---|---|---|---|
| Individual | I1 | | 0.580 |
| | I2 | | −0.04 * |
| | I3 | | −0.022 * |
| | I4 | | −0.037 * |
| | I5 | | −0.026 * |
| | I6 | | 0.015 |
| | I7 | | −0.077 * |
| | I8 | | −0.045 * |
| | I9 | | −0.048 * |
| | I10 | −0.053 * | −0.096 * |
| | I11 | | −0.099 * |
| | I12 | | −0.094 * |
| | I13 | | −0.088 * |
| | I14 | | −0.085 * |
| | I15 | | −0.113 * |
| | I16 | | −0.097 * |
| | I17 | | −0.092 * |
| | I18 | | −0.103 * |
| | I19 | | −0.05 * |
| Socio-economic | S1 | | 0.575 |
| | S2 | | 0.439 |
| | S3 | | 0.514 |
| | S4 | | 0.466 |
| | S5 | | 0.483 |
| | S6 | | 0.523 |
| | S7 | | 0.483 |
| | S8 | 0.530 | 0.484 |
| | S9 | | 0.492 |
| | S10 | | 0.497 |
| | S11 | | 0.542 |
| | S12 | | 0.548 |
| | S13 | | 0.453 |
| | S14 | | 0.609 |
| | S15 | | 0.526 |
| Academic | A1 | | 0.684 |
| | A2 | | 0.678 |
| | A3 | | 0.689 |
| | A4 | | 0.740 |
| | A5 | | 0.633 |
| | A6 | | 0.642 |
| | A7 | | 0.626 |
| | A8 | | 0.636 |
| | A9 | 0.670 | 0.642 |
| | A10 | | 0.643 |
| | A11 | | 0.651 |
| | A12 | | 0.635 |
| | A13 | | 0.660 |
| | A14 | | 0.654 |
| | A15 | | 0.701 |
| | A16 | | 0.664 |
| | A17 | | 0.663 |
| Institutional | IES1 | | 0.781 |
| | IES2 | | 0.680 |
| | IES3 | | 0.677 |
| | IES4 | | 0.713 |
| | IES5 | 0.744 | 0.720 |
| | IES6 | | 0.694 |
| | IES7 | | 0.698 |
| | IES8 | | 0.759 |

* The value is negative due to a negative average covariance between elements. These breach the assumptions of the reliability model; however, by removing some element, this value may fit the reliability model. ** $\alpha$-SE corresponds to the value of $\alpha$ if the element is removed.

*3.3. Data Analysis and Modelling*

With the data collected, due to the nature of the data and the purpose of this study, where rural students are categorised, we proceeded to identify the variables that influence the decision to drop out or stay in higher education. For this purpose, the Mann–Whitney U test was used because the data did not fit a normal distribution (see Table 4), and it facilitated the comparison of independent populations, in this case the students who expressed the intention to drop out or to stay in the training programme. The existence of statistically significant differences between the two groups of students for the study variables was present when the *p*-value was less than 0.05 [78].

**Table 4.** Kolmogorov–Smirnov normality test.

| Code | Statistic * | *p*-Value ** | Code | Statistic * | *p*-Value ** |
|---|---|---|---|---|---|
| I2 | 0.388 | <0.01 | S9 | 0.461 | <0.01 |
| I3 | 0.394 | <0.01 | S10 | 0.369 | <0.01 |
| I4 | 0.486 | <0.01 | S12 | 0.468 | <0.01 |
| I5 | 0.371 | <0.01 | S13 | 0.179 | <0.01 |
| I6 | 0.361 | <0.01 | S15 | 0.523 | <0.01 |
| I7 | 0.338 | <0.01 | A1 | 0.489 | <0.01 |
| I8 | 0.248 | <0.01 | A3 | 0.411 | <0.01 |
| I9 | 0.296 | <0.01 | A5 | 0.240 | <0.01 |
| I10 | 0.422 | <0.01 | A6 | 0.229 | <0.01 |
| I11 | 0.410 | <0.01 | A7 | 0.242 | <0.01 |
| I12 | 0.385 | <0.01 | A8 | 0.238 | <0.01 |
| I13 | 0.188 | <0.01 | A9 | 0.238 | <0.01 |
| I14 | 0.201 | <0.01 | A10 | 0.229 | <0.01 |
| I15 | 0.218 | <0.01 | A11 | 0.253 | <0.01 |
| I16 | 0.461 | <0.01 | A12 | 0.266 | <0.01 |
| I17 | 0.434 | <0.01 | A13 | 0.228 | <0.01 |
| I18 | 0.233 | <0.01 | A14 | 0.322 | <0.01 |
| I19 | 0.224 | <0.01 | A16 | 0.272 | <0.01 |
| S2 | 0.243 | <0.01 | A17 | 0.241 | <0.01 |
| S3 | 0.540 | <0.01 | IES2 | 0.257 | <0.01 |
| S4 | 0.472 | <0.01 | IES3 | 0.326 | <0.01 |
| S5 | 0.470 | <0.01 | IES4 | 0.467 | <0.01 |
| S6 | 0.540 | <0.01 | IES5 | 0.422 | <0.01 |
| S7 | 0.439 | <0.01 | IES6 | 0.365 | <0.01 |
| S8 | 0.474 | <0.01 | IES7 | 0.403 | <0.01 |

* The degrees of freedom (gl) were 269. ** Normal distribution is rejected with *p*-value < 0.05.

With the explanatory variables in which statistically significant differences were identified, we proceeded to compare the way in which the groups behaved in relation to these variables, therefore modelling based on clusters or classification was chosen, since this allows the description of groups with homogeneous characteristics based on the study variables of a particular event or phenomenon [79]. In this sense, cluster modelling assumes that individuals share a common distribution of characteristics, while different individuals follow a different distribution [80]. That is, a study population has a finite number of n distributions, and the purpose of clustering is to take such a mixture and analyse it into simple components and estimate the "membership probabilities" [79].

This type of modelling has both supervised and unsupervised techniques. Since there are no previous studies on the rural population to establish how students cluster, both those who wish to drop out and those who wish to stay in higher education, hierarchical cluster modelling was used. This type of modelling, being in the unsupervised category, does not require an underlying statistical model. Ward's technique was chosen to create the model because it minimises the sums of squares of each variable's deviations from the mean, allowing for homogenous groups of people. Furthermore, the squared Euclidean

distance interval was used to determine similarities and differences across observations, and data values were normalised to minimise the impacts of the questionnaire scales.

To establish differences between clusters, the Mann–Whitney U statistic was used if the number of clusters to be extracted was two, or, if the number was greater than two, the Kruskal–Wallis statistic was used. In either case, differences were considered statistically significant when the *p*-value was less than 0.05 [78]. Finally, descriptive statistics were used to identify the individual, socio-economic, academic and institutional characteristics that influence dropout and retention among rural students.

## 4. Results

Regarding the statistically significant differences between rural students with the intention to drop out or to remain in the higher education programme, it was identified that the explanatory variables I9, I15, S15, A12, A13, A14, A16, A17, IES2, IES3, IES5, IES6 and IES7 were those in which the participants in the sample differed from each other. Table 5 presents the results of the Mann–Whitney U test.

**Table 5.** Mann–Whitney U test results between students with intention to drop out and with intention to stay.

| Code | Statistic | *p*-Value * | Code | Statistic | *p*-Value * |
|------|-----------|-------------|------|-----------|-------------|
| I2 | 8577.500 | 0.395 | S9 | 8961.500 | 0.874 |
| I3 | 8841.500 | 0.717 | S10 | 8156.500 | 0.108 |
| I4 | 8399.000 | 0.156 | S12 | 8420.000 | 0.181 |
| I5 | 8238.500 | 0.145 | S13 | 8063.500 | 0.117 |
| I6 | 8308.000 | 0.194 | S15 | 8107.000 | 0.010 |
| I7 | 8585.500 | 0.432 | A1 | 8741.000 | 0.504 |
| I8 | 8056.500 | 0.103 | A3 | 8429.000 | 0.238 |
| I9 | 7801.500 | 0.038 | A5 | 8620.500 | 0.477 |
| I10 | 8692.000 | 0.473 | A6 | 8387.000 | 0.275 |
| I11 | 8361.000 | 0.171 | A7 | 8514.000 | 0.373 |
| I12 | 8775.000 | 0.611 | A8 | 8280.500 | 0.205 |
| I13 | 8341.500 | 0.260 | A9 | 8339.500 | 0.236 |
| I14 | 8102.500 | 0.128 | A10 | 8158.000 | 0.137 |
| I15 | 6905.000 | <0.01 | A11 | 8056.500 | 0.101 |
| I16 | 8246.000 | 0.064 | A12 | 7316.500 | 0.004 |
| I17 | 8536.500 | 0.283 | A13 | 7475.500 | 0.010 |
| I18 | 8715.500 | 0.602 | A14 | 7869.500 | 0.040 |
| I19 | 8862.000 | 0.773 | A16 | 6430.000 | 0.000 |
| S2 | 8924.500 | 0.850 | A17 | 7420.000 | 0.007 |
| S3 | 8932.500 | 0.684 | IES2 | 7189.500 | 0.002 |
| S4 | 8682.000 | 0.450 | IES3 | 6996.500 | 0.000 |
| S5 | 9020.000 | 0.968 | IES4 | 8232.500 | 0.087 |
| S6 | 8890.500 | 0.429 | IES5 | 7895.000 | 0.028 |
| S7 | 8848.000 | 0.708 | IES6 | 6862.000 | < 0.01 |
| S8 | 8658.000 | 0.418 | IES7 | 7638.000 | 0.009 |

* Difference of medians with *p*-value is accepted <0.05.

Taking the variables in which statistically significant differences were identified as a reference, it was found that the (male) parents of students with the intention of dropping out had a lower educational level. At the same time, this group of students most frequently expressed that work and family obligations reduced the time they spent on their education. The need to move to study in a place other than the place of origin was more frequent in the group of students with the intention of dropping out. In terms of academic performance, students who indicated their intention to stay considered their academic performance to be outstanding or excellent.

In the case of academic preparation at previous levels of education, students intending to drop out most frequently stated that they were not adequately prepared for higher education. In addition, there is a higher level of dissatisfaction in the choice of training

programmes among this student population, as well as a lack of access to technological resources for the correct development of their training programme.

However, for the variables of the institutional determinant, students with the intention of dropping out presented greater difficulties in communication with HEIs, as well as in attention from administrative staff. Similarly, this group of students consider that the technologies (e.g., virtual campus, specialised software and hardware) acquired by the institution are not necessarily the most appropriate, as they present greater dissatisfaction. The situation described above is the same in relation to their perception of the bibliographic resources (e.g., books or databases) that HEIs have. In relation to teaching, students with the intention of dropping out presented higher levels of dissatisfaction with the attention given by teachers to doubts and concerns, as well as the way in which the contents were taught. Table 6 presents the response counts for each of the student groups.

**Table 6.** Response counts among students with intention to drop out and to stay.

| Code | Options for Response | No * | Yes ** | No * | Yes ** |
|------|----------------------|------|--------|------|--------|
|      |                      | Count | | % | |
| I9 | Did not study | 18 | 17 | 13% | 13% |
|    | Primary | 56 | 74 | 41% | 56% |
|    | Secondary | 31 | 20 | 22% | 15% |
|    | Technical and technological | 8 | 7 | 6% | 5% |
|    | Professional | 15 | 3 | 11% | 2% |
|    | Postgraduate | 1 | 1 | 1% | 1% |
|    | Don't know | 9 | 9 | 7% | 7% |
|    | Total | 138 | 131 | 100% | 100% |
| I15 | Strongly disagree | 22 | 9 | 16% | 7% |
|     | Disagree | 33 | 16 | 24% | 12% |
|     | Neither disagree nor agree | 29 | 30 | 21% | 23% |
|     | Agree | 36 | 53 | 26% | 40% |
|     | Strongly agree | 18 | 23 | 13% | 18% |
|     | Total | 138 | 131 | 100% | 100% |
| S15 | Yes | 10 | 23 | 7% | 18% |
|     | No | 128 | 108 | 93% | 82% |
|     | Total | 138 | 131 | 100% | 100% |
| A12 | Deficient | 2 | 0 | 1% | 0% |
|     | Insufficient | 3 | 6 | 2% | 5% |
|     | Acceptable | 30 | 45 | 22% | 34% |
|     | Outstanding | 68 | 63 | 49% | 48% |
|     | Excellent | 35 | 17 | 25% | 13% |
|     | Total | 138 | 131 | 100% | 100% |
| A13 | Strongly disagree | 4 | 13 | 3% | 10% |
|     | Disagree | 13 | 14 | 9% | 11% |
|     | Neither disagree nor agree | 36 | 43 | 26% | 33% |
|     | Agree | 59 | 43 | 43% | 33% |
|     | Strongly agree | 26 | 18 | 19% | 14% |
|     | Total | 138 | 131 | 100% | 100% |

**Table 6.** *Cont.*

| Code | Options for Response | No * | Yes ** | No * | Yes ** |
|---|---|---|---|---|---|
| | | Count | | % | |
| A14 | Strongly disagree | 1 | 1 | 1% | 1% |
| | Disagree | 2 | 3 | 1% | 2% |
| | Neither disagree nor agree | 7 | 16 | 5% | 12% |
| | Agree | 46 | 47 | 33% | 36% |
| | Strongly agree | 82 | 64 | 59% | 49% |
| | Total | 138 | 131 | 100% | 100% |
| A16 | Strongly disagree | 2 | 6 | 1% | 5% |
| | Disagree | 6 | 14 | 4% | 11% |
| | Neither disagree nor agree | 11 | 17 | 8% | 13% |
| | Agree | 48 | 59 | 35% | 45% |
| | Strongly agree | 71 | 35 | 51% | 27% |
| | Total | 138 | 131 | 100% | 100% |
| A17 | Strongly disagree | 2 | 4 | 1% | 3% |
| | Disagree | 6 | 9 | 4% | 7% |
| | Neither disagree nor agree | 22 | 27 | 16% | 21% |
| | Agree | 48 | 55 | 35% | 42% |
| | Strongly agree | 60 | 36 | 43% | 27% |
| | Total | 138 | 131 | 100% | 100% |
| IES2 | Never | 15 | 33 | 11% | 25% |
| | Occasionally | 69 | 64 | 50% | 49% |
| | Always | 54 | 34 | 39% | 26% |
| | Total | 138 | 131 | 100% | 100% |
| IES3 | Never | 5 | 17 | 4% | 13% |
| | Occasionally | 49 | 60 | 36% | 46% |
| | Always | 84 | 54 | 61% | 41% |
| | Total | 138 | 131 | 100% | 100% |
| IES4 | Never | 1 | 3 | 1% | 2% |
| | Occasionally | 26 | 34 | 19% | 26% |
| | Always | 111 | 94 | 80% | 72% |
| | Total | 138 | 131 | 100% | 100% |
| IES5 | Never | 1 | 4 | 1% | 3% |
| | Occasionally | 36 | 47 | 26% | 36% |
| | Always | 101 | 80 | 73% | 61% |
| | Total | 138 | 131 | 100% | 100% |
| IES6 | Never | 1 | 12 | 1% | 9% |
| | Occasionally | 43 | 58 | 31% | 44% |
| | Always | 94 | 61 | 68% | 47% |
| | Total | 138 | 131 | 100% | 100% |
| IES7 | Never | 3 | 9 | 2% | 7% |
| | Occasionally | 36 | 47 | 26% | 36% |
| | Always | 99 | 75 | 72% | 57% |
| | Total | 138 | 131 | 100% | 100% |

* No, these are students who intend to stay. ** Yes, these are students with the intention to drop out. The variable codes are presented in Table A1.

### 4.1. Dropout in Rural Higher Education

With the explanatory variables for dropout and permanence in which statistically significant differences were identified, for the study population that reported wanting to drop out, we proceeded to develop the cluster-based model. Thus, the total of 131 cases that made up the sample section were validated for the development of the hierarchical cluster. The cut-off was made at the rescaled distance 20 (see Figure 3), thus forming two clusters, the first with n = 45 (CD1) and the second with n = 86 (CD2).

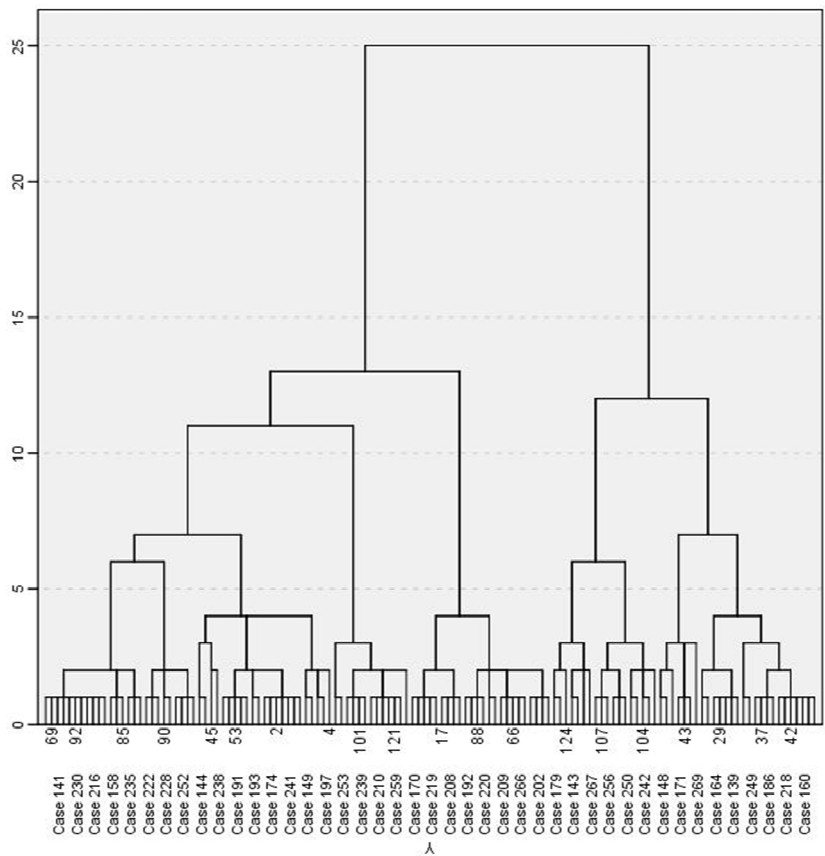

**Figure 3.** Dendrogram. Note: The *x*-axis represents the cases of students with intention to drop out, and the *y*-axis represents the combination of rescaled distance clusters.

Regarding statistically significant differences between clusters, the results of the Mann–Whitney U test are presented in Table 7. Differences were identified in the explanatory variables I9, I15, S15, A14, A16, IES2, IES3, IES4, IES5, IES6 and IES7.

Both clusters were characterised by low levels of parents' education. Thus, for CD1, 20% of its members reported that their father had no education at all, 62.2% had completed primary school, 11.1% had completed secondary school and only 2.2% had completed their undergraduate degree. CD2 members indicated that 9.3% had not completed any level of education, 53.5% had completed primary school, 17.4% had completed secondary school and 9.3% had completed an undergraduate degree, while 10.5% indicated that they did not know their own father.

**Table 7.** Mann–Whitney U test results for CD1 and CD2.

| Code | Statistic | *p*-Value * |
|------|-----------|------------|
| I9   | 1429.500  | 0.007      |
| I15  | 1419.000  | 0.009      |
| S15  | 1208.000  | <0.01      |
| A12  | 1769.000  | 0.382      |
| A13  | 1850.500  | 0.670      |
| A14  | 1175.000  | <0.01      |
| A16  | 1444.000  | 0.011      |
| A17  | 1724.000  | 0.280      |
| IES2 | 1139.000  | <0.01      |
| IES3 | 1135.000  | <0.01      |
| IES4 | 73.500    | <0.01      |
| IES5 | 899.000   | <0.01      |
| IES6 | 686.000   | <0.01      |
| IES7 | 1429.500  | 0.007      |

* Difference of medians with *p*-value < 0.05 is accepted.

Regarding work obligations, for CD1, 68.9% stated that these interfere with their educational process, while for CD2, the percentage was lower at 52.3%. In relation to the need to move from their place of origin to another city or municipality in order to be able to study, 42.2% of CD1 indicated having to do so. On the other hand, only 4.7% of CD2 students reported this situation. In the case of satisfaction with the choice of training programme, 66.7% of students in CD1 said they were satisfied, while 94.2% of students in CD2 said they were satisfied with their choice of training programme Concerning the availability of the necessary tools to carry out the work left in class (e.g., computer, internet, computer programs), 44.4% of the students in CD1 indicated that they did not have them, while only 9.3% of CD2 did not.

However, regarding the evaluation of the communication processes with HEIs, 88.9% of CD1 and 66.3% of CD2 indicated that it was not easy to communicate with HEIs. In line with the above, 82.2% of CD1 members perceive that HEI officials do not attend to their needs, and 46.5% of CD2 members perceive that HEI officials do not attend to their needs. In terms of the tools (e.g., databases, software, etc.) available to HEIs, 77.8% (CD1) and 18.6% (CD2) of students consider that these are not adequate. In relation to the training process, CD1 members tend to have perceptions that teachers do not deal with their doubts in a timely manner (84.4%), as well as that they do not impart the content in a simple way (82.2%). In these same aspects for CD2, 37.2% reported that teachers do not address their doubts, while 42.2% felt that they did not impart the content in a simple way.

*4.2. Permanence in Rural Higher Education*

In relation to the students who indicated that they wanted to remain in the undergraduate programme, it was identified that they conglomerate into two clusters (cut-off at the rescaled distance 20). Thus, the first cluster consisted of n = 99 (CP1) and the second of n = 39 (CP2). Figure 4 presents the dendrogram.

Regarding the statistically significant differences between clusters, Table 8 presents the results of the Mann–Whitney U test; differences were identified in the explanatory variables A12, A13, A14, A16, IES2, IES3, IES4, IES5, IES6 and IES7.

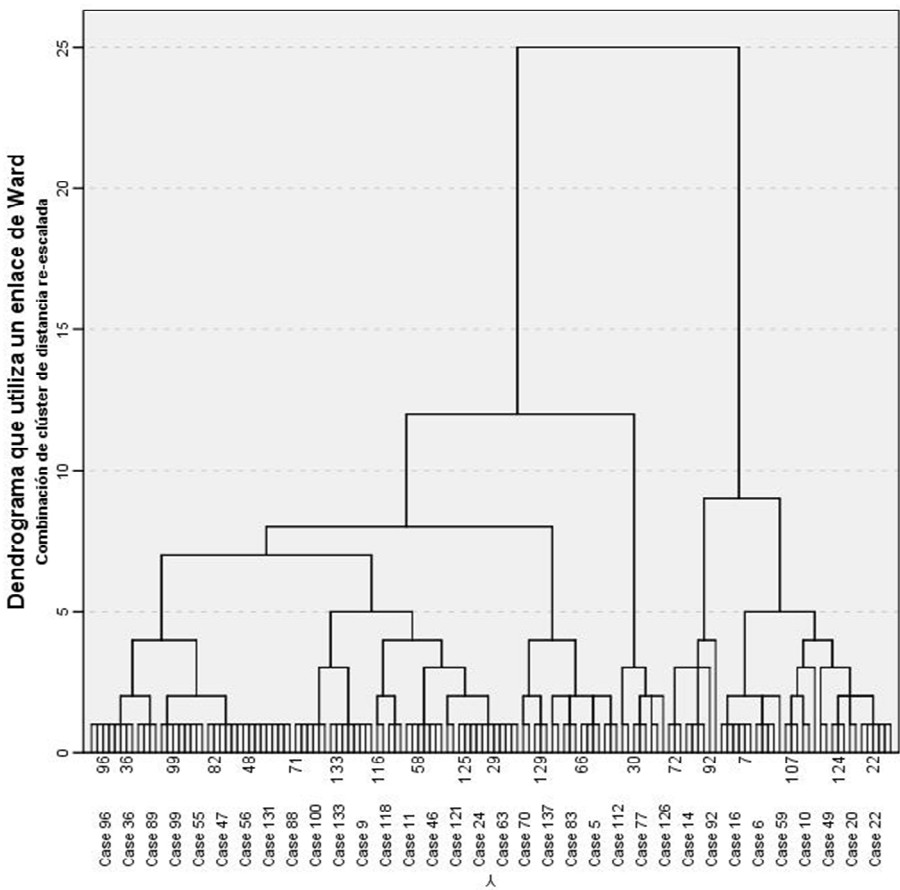

**Figure 4.** Dendrogram. Note: The *x*-axis represents the cases of students with intention to stay, and the *y*-axis represents the combination of rescaled distance clusters.

**Table 8.** Mann–Whitney U test results for CP1 and CP2.

| Code | Statistic | *p*-Value * |
|------|-----------|-------------|
| I9 | 1,827,000 | 0.609 |
| I15 | 1,704,500 | 0.274 |
| S15 | 1,873,500 | 0.548 |
| A12 | 1,536,500 | 0.044 |
| A13 | 1,449,000 | 0.016 |
| A14 | 1,206,500 | <0.01 |
| A16 | 1.547,000 | 0.045 |
| A17 | 1,563,500 | 0.063 |
| IES2 | 912,000 | <0.01 |
| IES3 | 920,000 | <0.01 |
| IES4 | 1,210,500 | <0.01 |
| IES5 | 852,500 | <0.01 |
| IES6 | 437,500 | <0.01 |
| IES7 | 675,000 | <0.01 |

* Difference of medians is accepted with *p*-value < 0.05.

In relation to the differences identified, 71.7% of the related students in CP1 indicated that they considered their GPA to be outstanding or excellent, while for CP2 it was 82.1%. Regarding the perception of the students' preparation for entry to higher education, 65.7% of CP1 stated that their teachers had prepared them adequately. In the case of CP2, only 52.3% considered that their teachers had prepared them adequately for entry to HEI. At the same time, CP1 students reported being satisfied with the choice of the training programme in which they are enrolled, while for CP2 only 74.4% were satisfied with the training

programme. Finally, 81.8% of CP1 and 69.2% of CP2 considered that they carry out their training activities on time.

However, in the case of the institutional explanatory variables, 49.5% of CP1 and 89.7% of CP2 stated that it was never or occasionally easy to communicate with the HEI. Consequently, 74.4% of CP2 stated that HEI administrative staff never or occasionally attend to their requests. On the other hand, 74.1% of CP1 assessed that the administrative staff of the HEIs did attend to their requests and concerns. As for the technologies (e.g., virtual campus, specialised software and hardware) used by the HEI where they are studying, 90.9% of CP1 and 53.8% of CP2 considered them adequate. Regarding the bibliographic resources (e.g., books or databases) held by HEIs, 88.9% of CP1 members considered them to be relevant for the development of their academic activities, while for CP2, 66.7% did not consider them to be appropriate. Finally, regarding institutional processes related to teachers, 89.9% of CP1 and 12.8% of CP2 reported that teachers dealt with their doubts and concerns in a timely manner. Similarly, in regard to the way in which teachers teach the contents of the subjects, the perception of 89.9% of the members of CP1 was positive, however, for CP2, only 25.6% agreed with it.

## 5. Discussion and Conclusions

As presented in the results section, it was found that student permanence and dropout in the rural population is influenced by different explanatory variables. Thus, the findings of the present study concerning the individual determinant identified, firstly, that dropout is related to the educational level of the father, which is a discrepancy with previous studies, such as those of Guzmán et al. [11], Barbosa-Camargo et al. [81] and Lundetræ [82], because dropout is usually related to the educational level of the mother. This may be the result of the influence of other variables not evaluated in the present study, such as the cultural factors of these populations, or the low inclusion rates of this gender in the educational system [83]. Secondly, work and family obligations make it difficult for rural students to remain in the education system, which is consistent with the study by Hart and Vender [28], who reported the relevance of this variable as a predictor of dropout in the rural student population.

In the case of the socio-economic determinant, it was established that there are no statistically significant differences in variables (type of housing, socio-economic stratum, access to public services, being a beneficiary of state subsidies, income level and financing of studies) that are traditionally conclusive in the dropout or permanence of other types of students, such as urban students [9,27]. In view of the above, it was found that students with the intention of dropping out most frequently expressed the need to move from their place of residence to pursue their education.

For the academic determinant variables, it was established that permanence in higher education is related to education at previous levels, coinciding with the studies of Choi and Park [18]. Similarly, the satisfaction of rural students with the training programme is a deterrent to the intention to drop out. Unlike the study developed by Guzmán et al. [11] for a rural student population in undergraduate programmes in virtual mode, in the present study, by linking the face-to-face mode, it was observed that the academic variables do have an impact on the events of desertion and permanence. Finally, in the explanatory variables of the institutional determinant only, no statistically significant differences were detected between the typology of students in relation to institutional welfare plans and extracurricular activities. The above is contrary to the results presented by Warner [84] and Nishat et al. [70].

With regard to the internal comparison between the groups of students (with the intention of leaving or remaining), it was determined that, in the case of CD1, this is associated with lower educational levels of the father, greater interference of work and family obligations with studying and lower evaluations with regard to having tools for the development of their work, satisfaction with the training programme, the ease of communication and attention with the HEI, the technologies and resources provided by the

institutions, the attention of the teachers and the simplicity of class teaching. For CD2, the main associated characteristic was the need to move from their place of origin to pursue their studies. However, in the case of permanence, when analysing the clusters, it was identified that there is no incidence of individual and socio-economic variables in this event, contrary to the findings of Georg [40], Guzmán et al. [85] and NEM [38]. CP1 was characterised by higher self-perceptions in relation to institutional variables, while CP2 was characterised by higher self-perceptions in relation to academic variables.

In light of what has been stated, it should be noted that the present study provides new insights into the events of dropout and retention in higher education for rural students by addressing variables that had not been previously addressed, such as marital status, type of housing, access to public services, state benefits, methods of financing their studies, dropout in other previous academic programmes, time of entry to higher education, number of subjects taken, attention from HEI administrative staff, the role of the teacher and extracurricular activities. The findings provide an opportunity for academics to further study these events and for public and institutional policy makers to modify current policies and create new ones in order to mitigate and prevent students' dropout, and consolidate their permanence at the educational level.

As a result of what has been described, it is necessary to recognise that policies that transcend over time are needed, with the aim of ensuring educational quality, reducing dropout indicators and increasing permanence rates in the rural student population. Considering that some of the variables that must be addressed for this purpose are not modifiable in the short or medium term, state efforts are required to improve the educational levels of parents, reduce the pressure of family and work obligations on students, improve academic performance prior to higher education, and support HEIs in adapting education to rural areas, especially when technologies are not adequate for this student population, among other factors.

To continue advancing in the research on dropout in rural higher education, it is necessary to verify how the variables studied behave in other contexts, both in developing and developed countries, in order to continue the discussion of the findings presented here. On the other hand, based on the modelling, it is necessary to verify the causal relationships between the determinants that explain the intentionality of permanence or dropout, in order to be able to propose strategies to prevent and mitigate student dropout in rural areas. On the other hand, the academic community is invited to continue deepening the study of this educational problem in rural populations, since it is not known what other variables analysed in other scenarios could influence the decision to stay or drop out.

However, for future research, some limitations of the methodological structure must be overcome, such as the transversality of the article, the sample size of the analysis groups, among other things. In addition, the results of the study must be interpreted from the limitations of the statistical analyses developed and the modelling technique selected, considering that all the variables analysed have the potential to explain dropout or retention in rural students studying in higher education; however, the variables that showed statistically significant differences are catalysts of these educational events. Finally, some of the findings presented here require further study, such as the limited influence of individual, socio-economic and academic variables on dropout or retention in rural higher education; hence, it is imperative to establish and deepen the causes of this absence.

**Author Contributions:** Conceptualization, A.G. and S.B.; methodology, A.G. and S.B.; software, A.G. and S.B.; validation, A.G. and S.B.; formal analysis, A.G. and S.B.; investigation, A.G. and S.B.; resources, A.G. and S.B.; data curation, A.G.; writing—original draft preparation, A.G. and S.B.; writing—review and editing, A.G., S.B. and F.C.-V.; visualization, A.G. and S.B.; supervision, S.B. and F.C.-V.; project administration A.G. and S.B. All authors have read and agreed to the published version of the manuscript.

**Funding:** This research received no external funding.

**Institutional Review Board Statement:** The study was conducted in accordance with the Declaration of Helsinki and approved by the Institutional Review Board of University Corporation of Asturias.

**Informed Consent Statement:** Informed consent was obtained from all subjects involved in the study.

**Data Availability Statement:** The data presented in this study are available on request from the corresponding author. The data are not publicly available due to the current Colombian laws that require the signing of a data transfer contract between the University Corporation of Asturias and the applicants.

**Acknowledgments:** The University Corporation of Asturias, whose support covered the cost of the publication, and Cecilia Carabajal, who, with her unconditional support, made the style corrections and translation of this article.

**Conflicts of Interest:** The authors declare no conflict of interest.

## Appendix A

**Table A1.** Self-report questionnaire.

| Code | Question | Options for Response | |
|------|----------|---------------------|---|
| I1 | Year of birth | | |
| I2 | What is your gender? | a. | Female. |
| | | b. | Male. |
| | | c. | Intersex. |
| | | d. | I prefer not to report. |
| I3 | At present, do you? | a. | Work full-time (48 h). |
| | | b. | Work part-time (from 20 to 24 h). |
| | | c. | Occasionally work (from 1 to 19 h). |
| | | d. | You are unemployed. |
| | | e. | You do not have the need to work. |
| I4 | Are you primarily responsible for your household expenses? | a. | Yes. |
| | | b. | No. |
| I5 | Do you have children under the age of 18? | a. | Yes. |
| | | b. | No. |
| I6 | Are you the person responsible for the upbringing of your children? | a. | Yes. |
| | | b. | No. |
| | | c. | Not applicable (Only if you answered no to question 5). |
| I7 | What is your marital status? | a. | Single (includes widowed, widower, divorced or separated). |
| | | b. | Married. |
| | | c. | In a common-law or de facto marital union. |

**Table A1.** *Cont.*

| Code | Question | Options for Response | |
|---|---|---|---|
| I8 | What is the highest level of education achieved by your mother? | a. <br> b. <br> c. <br> d. <br> e. <br> f. <br> g. | She did not study. <br> Primary school. <br> High School. <br> Technical and technological. <br> Vocational. <br> Postgraduate. <br> You had no relationship with your mother. |
| I9 | What is the highest level of education achieved by your father? | a. <br> b. <br> c. <br> d. <br> e. <br> f. <br> g. | He did not study. <br> Primary school <br> High School. <br> Technical and technological. <br> Vocational. <br> Postgraduate <br> You had no relationship with your father |
| I10 | I like studying | | |
| I11 | I feel that I am qualified to study at higher education level. | | |
| I12 | I am a responsible person for the execution of academic work independently. | | |
| I13 | I am frequently stressed by studying. | | |
| I14 | I feel that my family constantly interferes with my studies. | a. <br> b. <br> c. <br> d. <br> e. | Strongly disagree <br> Disagree <br> Neither disagree nor agree <br> Agree <br> Strongly agree |
| I15 | I feel that work or family obligations diminish the time I can devote to studying. | | |
| I16 | I am committed to the goal of completing my training programme. | | |
| I17 | I feel motivated to learn new concepts, themes and methodologies. | | |
| I18 | I am afraid of failing in a job, assignment and training programme. | | |
| I19 | I tend to procrastinate (leave everything to the last minute) in my daily activities, including my study. | | |
| S1 | The dwelling in which you live is. | a. <br> b. <br> c. <br> d. | Owned (you are the owner). <br> Family-owned (someone in your family owns it). <br> Leased. <br> Other type, which? |

**Table A1.** *Cont.*

| Code | Question | Options for Response |
|---|---|---|
| S2 | The house is in the stratum. | a. 1<br>b. 2<br>c. 3<br>d. 4<br>e. 5<br>f. 6<br>g. Don't know. |
| S3–S11 | The dwelling currently has access to the following services (multiple choice). | a. Water.<br>b. Sewerage.<br>c. Garbage collection.<br>d. Electricity.<br>e. Natural Gas.<br>f. Internet.<br>g. Landline.<br>h. Pay-TV service (satellite dish, cable, satellite, etc.). |
| S12 | Do you currently receive any benefits (e.g., education, health and transport) for being registered in SISBEN? | a. Yes<br>b. No<br>c. Don't know. |
| S13 | Does your family receive any state subsidy (Familias en Acción, Ingreso Seguro, Plan de Apoyo a la Vejez, etc.)? | a. Yes<br>b. No<br>c. Don't know. |
| S14 | Your family's income is between? | a. COP0 to COP500,000.<br>b. COP500,001 to COP1,000,000.<br>c. COP1,000,001 to COP1,500,000.<br>d. COP1,500,001 to COP2,000,000.<br>e. COP2,000,001 to COP2,500,000<br>f. COP2,500,000 or more. |
| S15 | Are your studies mainly funded by? | a. My income.<br>b. Parents.<br>c. Relatives other than parents (e.g., siblings, spouse, etc.)<br>d. Scholarships given by the Higher Education Institution or University.<br>e. Bank credit.<br>f. ICETEX Credit.<br>g. State programmes (e.g., ser pilo paga or generación E).<br>h. The university or Higher Education Institution is public and has no tuition fees.<br>i. Other source of funding, which? |
| S16 | Do you have to commute from your place of origin to another city to be able to study? | a. Yes.<br>b. No. |

**Table A1.** *Cont.*

| Code | Question | Options for Response |
|------|----------|----------------------|
| A1 | The secondary or high school from which you graduated was. | a. Official or public.<br>b. Private. |
| A2 | Prior to entering the training programme (technical, technological or vocational), you obtained information (e.g., curriculum, funding programme costs) to make the decision to enrol. | a. Yes.<br>b. No. |
| A3 | How much time passed between the enrolment to the undergraduate training programme (technical, technological or vocational) and the completion of your secondary school or high school? | a. Fewer than 6 months.<br>b. From 6 months to a year.<br>c. From 1 to 2 years.<br>d. From 2 to 3 years.<br>e. More than 3 years. |
| A4 | How many subjects do you take on average per academic semester? | a. 1<br>b. 2<br>c. 3<br>d. 4<br>e. 5<br>f. 6<br>g. 7<br>h. 8<br>i. More than eight. |
| A5 | Your performance during high school was: | |
| A6 | Your performance in the subject of Maths during high school was: | |
| A7 | Your performance in the subjects of the Natural Sciences during high school was: | |
| A8 | Your performance in the subject of Chemistry during the high school was: | a. Deficient<br>b. Insufficient<br>c. Acceptable<br>d. Outstanding<br>e. Excellent |
| A9 | Your performance in the subjects of Human Sciences (History, Geography, Philosophy, etc.) during high school was: | |
| A10 | Your performance in the subject of Spanish during high school was: | |
| A11 | Your performance in the subject of English during high school was: | |
| | You consider that your academic performance (average) during the time you have been linked to the Higher Education Institution or university has been. | |

**Table A1.** *Cont.*

| Code | Question | Options for Response |
|------|----------|----------------------|
| A12* | Your teachers have prepared you well for university. | a. Strongly disagree<br>b. Disagree<br>c. Neither disagree nor agree<br>d. Agree<br>e. Strongly agree |
| A13 | Your choice of undergraduate programme has satisfied you. | |
| A14 | The teachers in your degree programme often leave a lot of work. | |
| A15 | You have the necessary tools to do the work left in class (e.g., computer, internet, software). | |
| A16 | You hand in work left by the teacher on time. | |
| IES1 | How often have you made use of tutoring, psychological counselling, nutritional benefits and other programmes offered by your Higher Education Institution or University. | a. Never.<br>b. Occasionally.<br>c. Always. |
| IES2 | You considered it easy to communicate with the HEI/University through the channels defined by the HEI/University. | |
| IES3 | The administrative staff of the Higher Education Institution or University attended to their requirements. | |
| IES4 | The technologies (e.g., virtual campus, specialised software and hardware) used by the HEI or University were adequate for their training process. | |
| IES5 | The bibliographic resources (e.g., books or databases) owned by the HEI or university were relevant to the development of its academic activities. | |
| IES6 | Teachers tended to address their doubts and concerns in a timely manner. | |
| IES7 | Teachers taught the content of the subject in a simple way. | |
| IES8 | You were involved in extracurricular activities such as dance, sports, music, etc. | |

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
