# Peer review of "Comparative Analysis of Dropout and Student Permanence in Rural Higher Education"

_sustainability, doi:10.3390/su14148871_

Round 1

Reviewer 1 Report

Thank you very much for inviting me to review the paper entitled "Comparative Analysis of Dropout and Student Permanence in Rural Higher Education". the paper discusses issues with regards to rural students dropouts - using cluster modelling evidence were noted on top reasons for dropouts - i believed this paper is quite interesting and implications of findings significant.

Paper is also well written. Statistically sound.

Some minor recommendations are provided:

line 16 - abstract - aca-demic - should be academic, please recheck paper for similar issues, thanks

with regards to non-probabilistic - this should be further explain within the method section - can include how participants are recruited, from how many institutions, which geographical location? 

table 1 - income should be converted to USD or conversion rate should be provided including the date, so as to let readers get an idea of how much is 1500 pesos

take note of also the usage of "." decimal point, "," should be used for the thousand value

for the discussion - this can be expanded - besides noting previous studies for similar and contradicting findings, authors could focus further implication (practical and constructive), what now? what education providers are able to do? policy providers? although authors did provide some, however, improvement of parents' education would be quite a challenge, are there any other practical approaches?

in sum, the paper is quite interesting, just some minor revisions are needed.

Author Response

Point 1: line 16 - abstract - aca-demic - should be academic, please recheck paper for similar issues, thanks.

Response 1: A general review of typing errors in the manuscript was carried out.

Point 2: with regards to non-probabilistic - this should be further explain within the method section - can include how participants are recruited, from how many institutions, which geographical location?

Response 2: This was incorporated in lines 222-231, as well as in lines 236-237.

Point 3:  table 1 - income should be converted to USD or conversion rate should be provided including the date, so as to let readers get an idea of how much is 1500 pesos

Response 3: This was incorporated in lines 234-235.

Point 4:  for the discussion - this can be expanded - besides noting previous studies for similar and contradicting findings, authors could focus further implication (practical and constructive), what now? what education providers are able to do? policy providers? although authors did provide some, however, improvement of parents' education would be quite a challenge, are there any other practical approaches?

Response 3: This was incorporated in lines 490-499.

Reviewer 2 Report

Congratulations to the authors for the paper. Taking into account your instrument, the analytical tools and the formal way to solve the article, everything seems all right.

But if one take into account the topic and the discussion, some methodological doubts arise.

In the field of education, we are used to articles with  non-probabilistic and non-intentional samples. If you try to find out, for instance, that an innovative educational strategy has some benefits over a sample of students, that methodology is enough, at least as a proof of concept that can shed some light.

But in your case, this is not an educational study, it can be better considered as a sociological study. In this case, a non-probabilistic and non-intentional sampling is very weak. At least if you do not try to explain which part of the population can be represented by that sample, and how this has been achieved selecting population for your study. This has not been done or, at least, it does not clearly appear on you paper.

Taking into account that your sample can not be changed, at least you need to better explain which part of the Colombian population can be represented by that sample and how. Or, if this is not possible (and it seems so), you may explain that the sample is not representative anyhow. And, of course, you must limit your conclusions to this very narrow population example, avoiding to draw general conclusions (even in terms of "offering insights" 461-462). Those insights must be better contextualized to the sample scope.

You already point this methodological weakeness out on your last paragraph, which is remarkable. But I will suggest to better describe the sample and the methodological weakness at the beggining of the Methodology (215) in order to prevent the reader about the interpretation of data as representing any statistically relevant group of population.

Also, watch out a mistake between lines 138-139

Author Response

Point 1: But in your case, this is not an educational study, it can be better considered as a sociological study. In this case, a non-probabilistic and non-intentional sampling is very weak. At least if you do not try to explain which part of the population can be represented by that sample, and how this has been achieved selecting population for your study. This has not been done or, at least, it does not clearly appear on you paper.

Taking into account that your sample can not be changed, at least you need to better explain which part of the Colombian population can be represented by that sample and how. Or, if this is not possible (and it seems so), you may explain that the sample is not representative anyhow. And, of course, you must limit your conclusions to this very narrow population example, avoiding to draw general conclusions (even in terms of "offering insights" 461-462). Those insights must be better contextualized to the sample scope.

You already point this methodological weakeness out on your last paragraph, which is remarkable. But I will suggest to better describe the sample and the methodological weakness at the beggining of the Methodology (215) in order to prevent the reader about the interpretation of data as representing any statistically relevant group of population.

 Response 1: As mentioned, due to the type of sampling, it is not possible to generalise the conclusions presented, so the recommendation to place this limitation directly in the methodological section was accepted.

This manuscript is a resubmission of an earlier submission. The following is a list of the peer review reports and author responses from that submission.